# FedSHIBU: Federated Similarity-based Head Independent Body Update

Athul Sreemathy Raj [1], Irene Tenison [2,3], Kacem Khaled [1], Maroua Ben Attia [4], Felipe Gohring de Magalhães [1], Gabriela Nicolescu [1]

[1] Polytechnique Montréal , [2] Mila , [3] Université de Montréal , [4] Humanitas Solutions

## Abstract

Most federated learning algorithms like FedAVG aggregate client models to obtain a global model. However, this leads to loss of information, especially when the data distribution is highly heterogeneous across clients. As a motivation for this paper, we first show that data-specific global models (where the clients are grouped based on their data distribution) produce higher accuracy over FedAVG. This suggests a potential performance improvement if clients trained on similar data have a higher importance in model aggregation. We use data representations from extractors of client models to quantify data similarity. We propose using a weighted aggregation of client models where the weight is calculated based on the similarity of client data. Similar to FedBABU, the proposed *client representation similarity*-based aggregation is applied only on extractors. We empirically show that the proposed method enhances global model performance in heterogeneous data distributions.

## 1 Introduction

Federated learning is a distributed machine learning framework where several devices collectively train a model without accessing the data distributed across these devices. The devices or clients utilize the data collected or generated at their end, to train their local models. These local models are used to effectively train a global model housed in the server without direct access to the data. The most common algorithm to train the global model at the server in this setting is FedAVG [1], where the trained local model parameters are averaged to form the global model. FedAVG and similar aggregation strategies are particularly helpful when the distribution of data across the clients is homogeneous. However, the real-world data distributed across the clients are heterogeneous since they are dependent on user behavior and other client

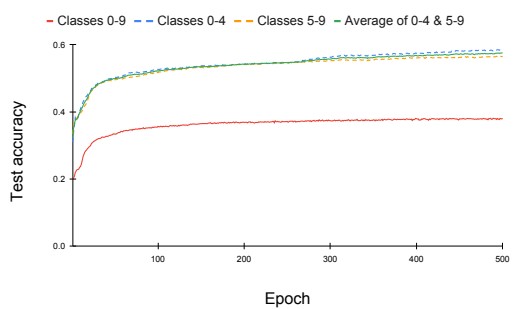

Figure 1: FL global models benefit from the grouping of clients based on their data distributions.

specifics. This is a major challenge in federated learning since it induces client-drift [2] which prevents global model convergence. Several researchers are focusing on enhancing the global model performance by effectively tackling the heterogeneity in data distribution [3–5]. Researchers have also been leveraging the data heterogeneity to enhance the personalization of the client models aiming at enhancing the client experience while the collective behavior is retained. [6–9] focus on building

Workshop on Federated Learning: Recent Advances and New Challenges, in Conjunction with NeurIPS 2022 (FL-NeurIPS'22). This workshop does not have official proceedings and this paper is non-archival.

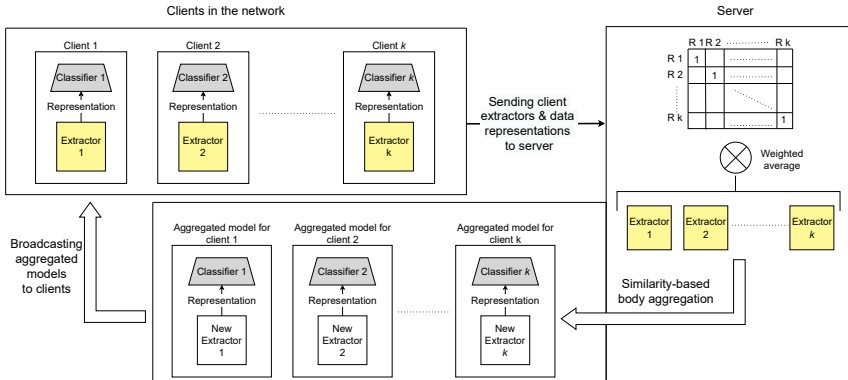

Figure 2: FedSHIBU - Extractors from individual clients are sent to the server, where the weighted average of models occurs. Afterward, the server broadcasts the updated models back to all clients.

multiple personalized models. FedBABU [10] focuses on building a global model capable of fitting to the local data of all clients. To enable this, FedBABU decouples the model into extractors and classifiers; the extractors are aggregated while the classifiers are only fine-tuned. The aggregated extractor and fine-tuned classifier together form the local model. They show that this enhances the personalization of client models since decoupling and fine-tuning protect classifiers from learning unnecessary and irrelevant information [10].

In this paper, we focus on extractors. We investigate whether the extractors are learning unnecessary information through aggregation and if this leads to performance degradation. To answer this question empirically, we simulate a simple federated setting with 10 clients. We distribute the CIFAR-10 dataset across these clients such that clients 0 to 4 each have examples from labels 0 to 4 and clients 5 to 9 each have examples from labels 5 to 9. In the first case, all extractors are aggregated during training. In the second case, clients 0 to 4 will have an extractor obtained by aggregating those from 0 to 4 only, and similarly, clients 5 to 9 will have an extractor obtained by aggregating those from 5 to 9 only. This is an explicit grouping of clients based on their data distribution. We run both experiments for 500 epochs and we observe that the average performance of all clients where the extractors were grouped is better than in the case where all client extractors are aggregated. The results are shown in Figure 1. This indicates that aggregation of client models having dissimilar or heterogeneous data distributions diminishes model performance, unlike cases when data distribution across clients is similar or homogeneous. This decrease in performance is due to information loss with model aggregation under heterogeneous data distribution [5].

Inspired by the above observation, we propose an algorithm that aggregates the extractors based on the similarity of data distributions across clients. This similarity-based aggregation is implemented on top of FedBABU. FedBABU training involves three stages - local client model updates, extractor aggregation at the server, and classifier fine-tuning at the clients. The proposed algorithm focus on extractor aggregation. A naive aggregation of extractors of all clients in the federated network is replaced by a weighted aggregation of other client extractors. The weighting is based on the similarity between a client's data distribution with the other clients in the network. We call this FedSHIBU, **Fed**erated **S**imilarity based **H**ead **I**ndependent **B**ody **U**pdate. The proposed algorithm has been summarized in Figure 2. Through similarity-based aggregation, FedSHIBU prioritizes client extractors trained on similar data distributions. This ensures that irrelevant information is not injected into the extractors. Our contributions are summarized as follows:

- We demonstrate that aggregation of all extractors as in FedBABU and other federated algorithms leads to client performance degradation and that selective aggregation based on data distribution likeliness diminishes this performance degradation.

- We propose a novel algorithm, **FedSHIBU**, which introduces a data distribution similarity-based aggregation as an alternative to aggregation of all extractors in FedBABU. A client representation similarity-matrix (CRSM) is calculated to quantify the similarity across data distributions and extractors are aggregated corresponding to their respective similarity score.

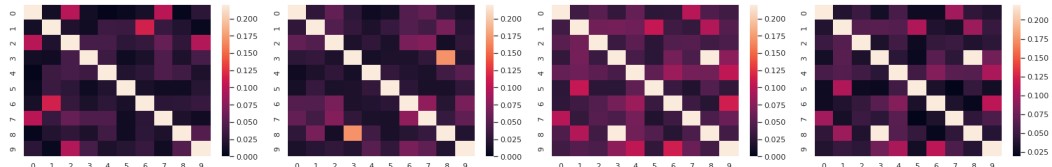

Figure 3: Similarity matrix from various phases of training using the proposed FedSHIBU algorithm. As training progresses we observe emerging patterns and groups and as it converges the groups become more prominent.

- We empirically show that similarity-based aggregation improves the performance of clients in both model-decoupled setting (FedBABU) and full-model setting (FedAVG).

The remainder of the paper is organized as follows: Section 2 reviews the background and the state-of-the-art that relates to our work; Section 3 introduces the basics of federated learning and FedBABU, along with our experiment setup; we explain our algorithm, experiments and obtained results inß Section 4; and Section 5 concludes this paper.

## 2    Related Works

**Federated Learning** proposed in [1] aims to learn a global model from multiple local models using decentralized data. They propose averaging the local model parameters to obtain the global model. Multiple iterations at the local model enable reduction of communication rounds required for convergence compared to FedSGD [1] where the local models are sent to the server after each iteration. However, multiple iterations cause client drift [2] when the data distribution across clients is heterogeneous. This bars the global model from converging to the optimal minima of the federated network. Several solutions have been proposed to handle this. [11] propose balancing the data distributions using data augmentation techniques to make it closer to an IID distribution. FedDyn [12] introduces a regularization term and FedProx [3] proposes a proximal term on the local objectives so as to penalize the clients diverging from the global model. Control variates were used in SCAFFOLD [2] to minimize the drift of local models from the global model. [13] aligns the features from the client networks to improve performance. FedNova [4] normalizes the gradients before averaging gradients and FedGMA [5] masks inconsistent gradients to enhance convergence of the global model. Personalized FL aims to make the local models cater to the specific data distribution at each client. Local models without federation are an alternative but every client may not have enough data to train their respective models. [14] clusters clients using unsupervised clustering algorithms on local client updates. Federated multi-task learning proposes task-based global models [15]. [16] uses transfer learning to transfer knowledge across clients. Regularizers are used by [17, 18] for personalization by preventing the models from being closer to global models. PerFedAVG [19] uses bi-level optimization.

**Representation Similarity Analysis (RSA)** is a data-analysis framework first introduced to correlate brain activities quantitatively in neuroscience [20]. This method uses pair-wise comparisons of data to reveal more information [21]. RSA methods in neuroscience use distance matrices to compute similarity [22]. It is widely used to analyze fMRI images of the brain and its specific regions of interest [23, 21] or to differentiate between stimuli [24]. It was later adopted to quantify the relationship between deep neural networks. In transfer learning and task taxonomy, RSA was used to quantify task similarity and cluster tasks [25]. It has been used to study the evolution of networks as training progresses [26]. Various matrices like CCA [26] and CKA [27] were developed by extending this principle. However, RSA is not devoid of pitfalls. In neuroscience, when the stimuli are confounding it tends to have higher RSA scores though they are from dissimilar systems. This is called the "mimic effect" [28] and leads to false inferences.

# 3 Preliminaries

## 3.1 Federated Learning

---

**Algorithm 1** FedSHIBU - adapted from FedBABU [10]

**function** FEDSHIBU
    initialize $\theta_G^0 = \{\theta_{G,ext}^0, \theta_{G,cls}^0\}$
    **for** each round $k = 1,...,K$ **do**
        $C^k \leftarrow$ random subset of $m$ clients
        **for** each client $C_i^k$ in parallel **do**
            $\theta_i^k(0) \leftarrow \theta_G^{k-1} = \{\theta_{G,ext}^{k-1}, \theta_{G,cls}^0\}$
            $\theta_{i,ext}^k, \hat{D}_i \leftarrow$ **ClientBodyUpdate**$(\theta_i^k(0), \tau)$
        **end for**
        $\theta_{G,ext,\{1,2,..m\}}^k \leftarrow$ **CRSMAggregation**$(\theta_{ext}^k, \hat{D})$
    **end for**
**return** $\theta_G^k = \{\theta_{G,ext}^K, \theta_{G,cls}^0\}$

---

**Algorithm 2** Updating body of client

**function** CLIENTBODYUPDATE$(\theta_i^k, \tau)$
    **for** each local epoch $1,...,\tau$ **do**
        $\theta_{i,ext}^k \leftarrow SGD(\theta_{i,ext}^k, \theta_{0,cls}^k)$
    **end for**
    $D_i \leftarrow$ random subset of data samples at client i
    $\hat{D}_i \leftarrow f(\theta_{i,ext}^k, D)$
**return** $\theta_{i,ext}^k, \hat{D}_i$

---

**Algorithm 3** Client Representation Similarity Matrix (CRSM) based Aggregation of extractors

**function** CRSMAGGREGATION$(\theta_{ext}^k, \hat{D})$
    **for** each client, $C_{cur}=1,...,m$ **do**
        **for** each client, $C_{rel}=1,...,m$ **do**
            $CRSM[C_{cur}, C_{rel}] \leftarrow$ **SimMet**$(\hat{D}_{C_{cur}}, \hat{D}_{C_{rel}})$
        **end for**
        $\theta_{G,ext,C_{cur}}^k \leftarrow \sum_{C_{rel}=1}^m \frac{CRSM[C_{cur},C_{rel}]}{\sum CRSM[C_{cur}]} \times \theta_{ext,C_{rel}}$
    **end for**
**return** $\theta_{G,ext,1,...,m}^k$

---

We summarize FL training procedure and notations used in Algorithm 1. Assume $1, , N$ is the set of all clients in the network. In every communication round , a random subset of $m$ clients are chosen to participate in training. The local model parameters of all participating clients $\theta^k(0)_{i=1}^m$ are initialized with the global model parameters $\theta_G^{k-1}$. $\theta_i^k(0) \leftarrow \theta_G^{k-1} \,\forall\, i = 1,..,m$. $\theta_i^k(0)$ is the local model parameters of client $i$ in communication round $k$ at local epoch $\tau = 0$. $\theta_G^0$ is randomly initialized for the first communication round, $k = 1$. Each client then updates its local models for $\tau$ iterations through their local data, $D_i$. The local model parameters, $\theta^k(\tau)$ from all clients are returned to the server where they are aggregated to obtain the global model $\theta_G^k$ for that communication round $k$; $\theta_G^k = \sum_{i=1}^m \theta_i^k(\tau)$. Our research focuses on a balanced data distribution. When the number of data samples per client varies, the local model parameters are weighted propotional their sample size; $\theta_G^k = \sum_{i=1}^m \frac{D_i}{\sum_{j=1}^m D_j} \theta_i^k(\tau)$.

## 3.2 FedBABU

In FedBABU [10], client models are decoupled into classifiers(head) $\theta_{cls}$ and extractors(body) $\theta_{ext}$. After training of local models at all participating clients, the extractor parameters are sent back to the server for aggregation. The extractors are aggregated $\theta_{G,ext}^k = \sum_{i=1}^m \frac{D_i}{\sum_{j=1}^m D_j} \theta_{i,ext}^k(\tau)$ leaving the classifiers unchanged. Decoupling of the network to extractors and classifiers helps reduce the bias in the classifiers in settings where the data distribution varies like class-imbalance settings [29]. The classifiers $\theta_{G,ext}^k$ are fine-tuned to enhance the personalization performance of the client models. For client updates in the ClientBodyUpdate function, the local extractor parameter $\theta_{i,ext}^k$ are updated based on the same classifier $\theta_{G,cls}^0$ such that the global parameters have the same classifier parameter as explained in Section 5.2 of [10]. It is to be noted that in FedBABU [29], all extractors are aggregated to form the global extractor which is passed over to all clients. We propose a client representation similarity matrix-based aggregation as given in the CRSMAggregation function of Algorithm 3 and further explained in Section 4 C.

## 3.3 Experimental Setup and Evaluation

ResNet18 has been used on CIFAR10 and CIFAR100 for all experiments. The separation of extractors and classifiers has been defined similarly to that in [29]. All convolutional layers including the pool layers in between form the extractor. That is, the representation returned in Algorithm 2 is the output of the last convolutional layer in the network. All dense layers following the extractor form the classifier. Heterogeneous distribution of data across clients also followed the pattern of [29] and [1]. $m$ is the number of participating clients in each communication round and we assume $m = N$ in our experiments. $s$ is the shards per user [1] and it determines the level of heterogeneity. A lower value of $s$ implies a higher heterogeneity in the data distribution. $\tau$ is the number of local epochs during client model training. A learning rate of 0.1 is used and it is decayed as in [29].

Table 1: Test Accuracy of FedSHIBU, FedBABU, and FedAVG on CIFAR100 and CIFAR10 distributed across 100 clients with full participation. The algorithms are compared across varying data heterogeneity where s=2 implies extreme heterogeneity and s=100 implies homogeneity.

| | CIFAR100 | | | | CIFAR10 | | | |
|---|---|---|---|---|---|---|---|---|
| s | FedSHIBU (CKA Linear) | FedSHIBU (CKA RBF) | FedBABU | FedAVG | FedSHIBU (CKA Linear) | FedSHIBU (CKA RBF) | FedBABU | FedAVG |
| 2 | **91.12** | 41.65 | 15.55 | 24.92 | **94.04** | 90.69 | 68.43 | 58.13 |
| 3 | **54.12** | 41.95 | 33.03 | 37.98 | **81.12** | 78.62 | 77.74 | 69.58 |
| 4 | **57.96** | 49.47 | 47.22 | 47.27 | 85.07 | **85.9** | 84.06 | 83.88 |
| 5 | **59.48** | 53.62 | 53.88 | 52.36 | **89.26** | 88.18 | 86.93 | 85.96 |
| 8 | **63.63** | 62.71 | 60.77 | 60.36 | 89.32 | **89.92** | 89.38 | 89.01 |
| 10 | **69.72** | 66.07 | 66.46 | 62.45 | **90.34** | 90.28 | 90.24 | 89.67 |
| 20 | **72.33** | 69.97 | 71.86 | 67.31 | 90.64 | **90.93** | 90.85 | 90.58 |
| 50 | **73.88** | 70.93 | 73.44 | 69.01 | 90.88 | 90.94 | 91.03 | **91.28** |
| 100 | 72.97 | 71.31 | **73.37** | 70.17 | 90.93 | **91.23** | 90.87 | 90.94 |

## 4 FedSHIBU - Federated Similarity-based Head Independent Body Update

FedSHIBU uses RSA to enhance the personalization of clients, by utilizing a data representation similarity matrix to weigh the clients based on their relative similarity.

### 4.1 FedBABU in heterogeneous data distribution

FedBABU averages all extractors to form a global model extractor. It then initializes the extractor of client models. This is particularly helpful when the data across clients are similar and the extractors retrieve features that are relevant to all clients. FedBABU performs better than FedAVG [1] when data distribution across clients is heterogeneous ($s > 5$). However, when the data is extremely heterogeneous ($s < 5$), the features learned by an extractor will be less relevant to classifiers of clients having distant data distributions. In these cases, FedBABU fails to outperform FedAVG. Heterogeneity in FedBABU [29] evaluation is limited to $s \geq 10$. From our experiments in Table 1, we observe that when $s < 5$ in CIFAR100, FedAVG outperforms FedBABU. However, the same was not observed in CIFAR10 which is a 10-class dataset, and the heterogeneity with $s < 5$ is not as severe as in CIFAR100 which is a 100-class dataset. We hypothesize that this is because of the presence of less-relevant extractors in the aggregation which results in an extractor with diminished abilities to extract relevant features. We propose to handle this by using a similarity based weighted aggregation of extractors.

### 4.2 Client Similarity

Representation Dissimilarity Matrix(RDM) used in [20] constitutes of $(1 - $Pearson's correlation$)$ of the pairwise conditions. In computer vision transfer learning [25], Spearman's correlation was used on these matrices to compute the similarity score of two DNNs. This was because RDM cannot be used directly since the comparison is across tasks and the shapes of representations across tasks will be different. However, in our federated scenario, all client models are assumed to have the same architecture. Hence the shape of representations from extractors is expected to be the same. To quantify the relationship between clients in federated learning, we propose using a client representation similarity matrix (CRSM). CRSM is a square matrix of size $N$x$N$, where each element is a pairwise similarity score of neural network

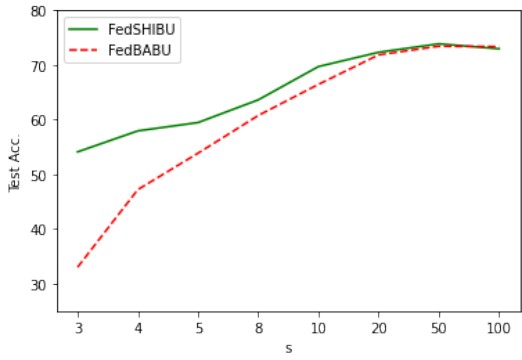

Figure 4: FedSHIBU outperforms FedBABU when data is extremely heterogeneous. With decreasing heterogeneity, the difference in performance decreases.

representations from client extractors. $CRSM[C_{cur}, C_{rel}]$ is the similarity of the current client $C_{cur}$ to another relative client $C_{rel}$. To measure the similarity score of neural network representations we use CKA linear and CKA RBF [27].

CRSM at various levels of training is shown in Figure 3. The matrices are from 10 client federated networks where the data distribution is heterogeneous ($s = 5$). We observe that initially, the grouping of clients changes very frequently. As training progresses and nears convergence, patterns are formed. The sub-figures in Figure 3 are from rounds 0, 150, 250, and 300 respectively. Until round 150, the values in CRSM varies drastically and in round 150 client 3 and 8 identifies each other as relatively similar to each other. As training progresses and reaches 200 rounds, the relevance is further strengthened and almost all clients have identified their relatives. With further training towards round 300, the gap between the groups are widened making the groups more prominent.

### 4.3 FedSHIBU algorithm

We propose a new FL algorithm called FedSHIBU (**Fed**erated **S**imilarity based **H**ead **I**ndependent **B**ody **U**pdate), an improvement upon the decoupled federated training strategy introduced in Fed-BABU [10]. By decoupling extractors and classifiers, only the body is trained while the head is never trained. This enhances the performance of client models since extractors hold information on data representations that are useful for all clients. Aggregating them enhances client performance. The classifiers hold information related to linear decision boundaries of clients and these are client data specific. Aggregating these leads to performance degradation. The algorithm and implementation are explained in detail in section 5.2 of [10].

Unlike FedBABU, extractors for each client in FedSHIBU are calculated at the server with a weighted aggregation of other client extractors in the network, where the weighting is dependent on the data representation similarities across clients. For the same, FedSHIBU requires client data representations to be sent to the server besides client updates. A fixed number of data samples are randomly chosen from each client and their representations from the extractors are retrieved after local training. Additional privacy mechanisms can be introduced to further enhance the privacy. The representations from clients are used to quantify the similarity of data across the clients in the network. The scores are collected as a matrix. A client extractor is obtained by aggregating other client extractors in the network with a weighting corresponding to their score in the matrix. This ensures that similar clients, having useful features gets weighed more in the aggregation than dissimilar clients, whose features may be less useful for the client model under consideration.

The training procedure of FedSHIBU is described in Algorithm 1. FedSHIBU introduces **CRSMAggregation** as given in Algorithm 3. CRSMAggregation function requires all client extractor parameters $\theta_{ext,1,2,..m}^{k}$ (also represented as $\theta_{ext}^{k}$) and all client representations from their extractors $\hat{D}_{1,2,..m}$ (also represented as $\hat{D}$). It calculates the similarity of representations to each other as a proxy of the similarity of data distributions at the clients. **SimMet** in Algorithm 3 can be any similarity metric. The similarities are compiled into a matrix - CRSM. The aggregated extractor for each client $C_{cur}$ is calculated by aggregating the client extractors by weighting them proportional to their similarity to other clients $C_{rel}$ in the federated network. This similarity is quantified in the corresponding row of CRSM $CRSM[C_{cur}]$ as given in Algorithm 3. Each client's aggregated extractors would differ since the weightage of different clients would vary in each aggregation.

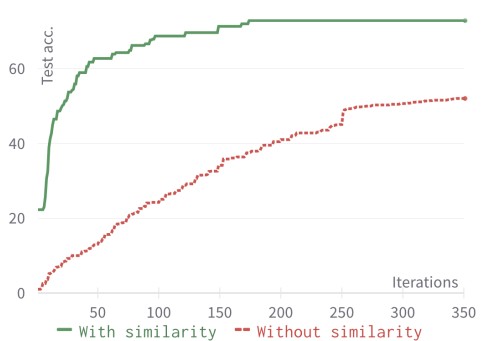

Figure 5: FedAVG with Similarity outperforms naive FedAVG under extremely heterogeneous data distributions.

FedSHIBU aims to handle extreme data heterogeneity, where FedBABU fails as observed in Table 1. Figure 4 plots test accuracy of FedSHIBU and FedBABU across varying heterogeneity represented

by $s$. When the data is heterogeneous, FedSHIBU outperforms FedBABU, and with increasing heterogeneity, the difference between the accuracy of FedSHIBU and FedBABU increases. That is, FedSHIBU is significantly better under extreme heterogeneity. This supports our hypothesis that FedSHIBU is capable of better handling information loss than FedBABU. When the data distribution is homogeneous, all clients would be weighed equally and FedSHIBU would equal FedBABU.

### 4.4 Similarity based aggregation in FedAVG

To understand the effect of representation similarity-based aggregation, we apply CRSMAggregation on full federated averaging without model decoupling. This helps understand the performance improvement contributed by similarity-based aggregation specifically. Figure 5 plots test accuracies of FedAVG and FedAVG with similarity-based aggregation on heterogeneous data ($s = 5$). We observe significant performance improvement along with faster convergence when similarity-based aggregation is employed. This further validates the claim of using similarity-based aggregation in federated learning. FedAVG with similarity-based aggregation is given in the Algorithm 4.

## 5 Conclusion

This work is an improvement of FedBABU[10] algorithm, by implementing an intelligent selection of clients for producing personalized client models. FedSHIBU converges faster than FedBABU and shows higher accuracy on client-specific datasets. However, calculating the similarity matrix is an overhead for the central node and costs compute. But since this activity is on a device with adequate resources, this wouldn't be a limiting factor in real-life scenarios. The generalization performance of FedSHIBU should be considered outside the scope of this work, and hence, is not bench-marked in this paper. It is true that this algorithm deviates from the core ideas of FL to have a global model. Again, since the target of this project is to improve personalization performance, it could be considered out of the scope of this work. In the future, we would like to experiment with more metrics to quantify client data similarity. We would also like to investigate the performance improvements of FedSHIBU in more complex datasets and distribution skews to further back the claims of this paper.

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

# A Appendix

---

**Algorithm 4** FedAVG with CRSM Aggregation

---

initialize $\theta_G^0 = \{\theta_{G,ext}^0, \theta_{G,cls}^0\}$
**for** each round $k = 1,...,K$ **do**
    $C^k \leftarrow$ random subset of m clients
    **for** each client $C_i^k$ in parallel **do**
        $\theta_i^k(0) \leftarrow \theta_G^{k-1} = \{\theta_{G,ext}^{k-1}, \theta_{G,cls}^0\}$
        $\theta_i^k, \hat{D}_i \leftarrow$ **ClientBodyUpdate**$(\theta_i^k(0), \tau)$
    **end for**
    $\theta_{G,\{1,2,..m\}}^k \leftarrow$ **CRSMAggregation**$(\theta^k, \hat{D})$
**end for**
**return** $\theta_G^k$
**function** CLIENTBODYUPDATE$(\theta_i^k, \tau)$
    **for** each local epoch $1,...,\tau$ **do**
        $\theta_i^k \leftarrow SGD(\theta_i^k)$
    **end for**
    $D_i \leftarrow$ random subset of data samples at client $i$
    $\hat{D}_i \leftarrow f(\theta_{i,ext}^k, D)$
**return** $\theta_i^k, \hat{D}_i$

---

