# OpenReview forum: "FedSHIBU: Federated Similarity-based Head Independent Body Update"
_NeurIPS.cc/2022/Workshop/Federated_Learning — FL-NeurIPS 2022 Poster_

### Official Review · Reviewer_296P · 2022-10-04
**The manuscript mainly focuses on the highly heterogeneous clients in federated learning. The authors use data representations from extractors of client models to quantify data similarity. A weighted aggregation of client models is proposed, in which the weight is calculated based on the similarity of client data.**

The manuscript mainly focuses on the highly heterogeneous clients in federated learning. The authors use data representations from extractors of client models to quantify data similarity. A weighted aggregation of client models is proposed, in which the weight is calculated based on the similarity of client data. Detailed comments are listed as follows.

1. There are some works that focus on weighted FL aggregation algorithms. It is recommended to compare them in related work.
2. It is recommended to quantify the similarity of clients in mathematical form.
3. It is suggested to put the part of the experimental setup and evaluation in Section 5 and put the conclusion in Section 6.
4. It is not convincing to use only CIFAR10 and CIFAR100 to verify highly heterogeneous clients.
5. The proposed weighted method based on data sample size is one-sided.

---

### Official Review · Reviewer_NUFB · 2022-10-17

This paper considers the personalization problem in federated learning. The authors propose a new scheme, FedSHIBU, to adapt to the data heterogeneity across devices. Similar to FedBABU, FedSHIBU decomposes the model into two parts: an extractor and a classifier, and only train and aggregate the extractor. During aggregation, each participant provides an additional representation vector of local data so that the server can compute the data-dependent similarity matrix. Then the server updates each device's model using its own weights.

- The algorithms is presented in a very confusing way. In Algorithm 1: $\theta_{G,ext}^k$ is not updated? Are we using a global model or not? In Algorithm 2: The function `f` is not defined? What is $\theta^k_{0,cls}$?

- The evaluation seems unfair. Does it look like FedSHIBU evaluates personalized models on their local distributions and then averages the accuracies? If so, then comparing with FedAvg is not fair. Indeed, compare with local training make more sense.

---

### Official Review · Reviewer_8w4H · 2022-10-18
**FL for heterogenous data sets**

This paper considers the federated learning problem. Most FL algorithms train a global model that works well over whole clients. However, when the data heterogeneity is severe, such FL algorithms suffer significant performance degradation. This paper does not consider making a single global model. Instead, the authors propose a weighted sum method that controls the local parameters without an aggregation step. The results outperform all the previous algorithms when data is heterogeneous.

Pros.

This paper is well-written and easy to follow.

The proposed algorithm considers an interesting new direction to resolve data heterogenous problems.

---

### Decision · Program_Chairs · 2022-10-20

Accept (Poster)